# Identification of Mycoses in Developing Countries

**DOI:** 10.3390/jof5040090

**Published:** 2019-09-29

**Authors:** Amir Arastehfar, Brian L. Wickes, Macit Ilkit, David H. Pincus, Farnaz Daneshnia, Weihua Pan, Wenjie Fang, Teun Boekhout

**Affiliations:** 1Westerdijk Fungal Biodiversity Institute, 3584 CT Utrecht, The Netherlands; f.daneshnia@wi.knaw.nl (F.D.); t.boekhout@wi.knaw.nl (T.B.); 2The Department of Microbiology, Immunology, and Molecular Genetics, The University of Texas Health Science Center at San Antonio, San Antonio, TX 78229, USA; wickes@uthscsa.edu; 3Division of Mycology, Department of Microbiology, Faculty of Medicine, University of Çukurova, Adana 01330, Turkey; macitilkit@gmail.com; 4bioMérieux, Inc., Hazelwood, MO 63042, USA; dave.pincus@biomerieux.com; 5Department of Dermatology, Shanghai Key Laboratory of Molecular Medical Mycology, Shanghai Institute of Medical Mycology, Second Military Medical University, Shanghai 200003, China; fangwenjie1990@126.com; 6Institute of Biodiversity and Ecosystem Dynamics, University of Amsterdam, 1012 WX Amsterdam, The Netherlands

**Keywords:** phenotypic assays, molecular tools, serology, Nanopore sequencing

## Abstract

Extensive advances in technology offer a vast variety of diagnostic methods that save time and costs, but identification of fungal species causing human infections remains challenging in developing countries. Since the echinocandins, antifungals widely used to treat invasive mycoses, are still unavailable in developing countries where a considerable number of problematic fungal species are present, rapid and reliable identification is of paramount importance. Unaffordability, large footprints, lack of skilled personnel, and high costs associated with maintenance and infrastructure are the main factors precluding the establishment of high-precision technologies that can replace inexpensive yet time-consuming and inaccurate phenotypic methods. In addition, point-of-care lateral flow assay tests are available for the diagnosis of *Aspergillus* and *Cryptococcus* and are highly relevant for developing countries. An *Aspergillus* galactomannan lateral flow assay is also now available. Real-time PCR remains difficult to standardize and is not widespread in countries with limited resources. Isothermal and conventional PCR-based amplification assays may be alternative solutions. The combination of real-time PCR and serological assays can significantly increase diagnostic efficiency. However, this approach is too expensive for medical institutions in developing countries. Further advances in next-generation sequencing and other innovative technologies such as clustered regularly interspaced short palindromic repeats (CRISPR)-based diagnostic tools may lead to efficient, alternate methods that can be used in point-of-care assays, which may supplement or replace some of the current technologies and improve the diagnostics of fungal infections in developing countries.

## 1. Introduction

Although fungal infections are generally neglected by healthcare communities, there is mounting evidence associating fungi with many pathological complications such as bowel-related diseases [1], neurodegenerative diseases [2], and invasive bloodstream infections [3]. Fungal infections cause 1.5 million deaths annually [4] and the number of patients prone to development of these infections is on the rise [5,6,7]. The lack of clinically effective antifungal agents together with the emergence of multi-drug-resistant organisms and new fungal species are major public health threats [8,9]. A broadening spectrum of fungal species causing infections in humans and species-specific susceptibility to antifungals underscore the importance of accurate identification to the species level and, for some fungi, sub-species level. However, reliable diagnostic tools are not always available in some countries. Thus, high-precision identification methods such as matrix-assisted laser desorption/ionization mass spectrometry (MALDI-TOF MS) and Sanger sequencing are not usually accessible in developing countries, where non-specific and time-consuming phenotypic assays are more commonly used [10,11]. A survey conducted by the Asian Fungal Working Group comprising seven South/Southeast Asian countries revealed that MALDI-TOF MS and sequencing are used only in 12–17% of the centers [12], among which almost 50% were referral hospitals from India and China. Although culture remains the gold standard diagnostic method in medical mycology, it lacks sensitivity and is time-consuming. Therefore, considering the importance of timely application of appropriate antifungal drugs for therapeutic success, the development of rapid and reliable diagnostic tools may have a significant influence on the reduction of the mortality rate [5]. Unfortunately, *Cryptococcus* infections in sub-Saharan countries [13], candidemia caused by *Candida auris* in India [8], most infections caused by dimorphic fungi [14], and eumycetoma [15] are among the most challenging to diagnose in developing and poor countries. Non-specific identification may contribute to the persistence of hard-to-eradicate fungal species such as *C*. *auris*, which may contribute to horizontal transfer subsequently resulting in clonal outbreaks [16].

In this review, we explore and discuss the application, advantages, and disadvantages of identification tools predominantly used in laboratories of developing countries, including phenotypic, molecular, and serological assays. Furthermore, because the number of antifungal-resistant species is increasing, we review the molecular methods used to supplement common antifungal susceptibility testing. Finally, we discuss new promising technologies that may be employed in future point-of-care diagnostic testing to detect fungal species in developing countries.

## 2. Phenotypic Assays

### 2.1. Blood Culture Bottles: Automated or Manual?

Culture as the gold standard technique is the cornerstone of patient management in clinical settings, especially for invasive fungal infection cases. Once isolated, a fungal agent can be subjected to downstream applications, such as identification, antifungal susceptibility testing, and typing techniques to trace the source of infection followed by implementing an appropriate infection control strategy and administration of proper antifungal therapy. As normal solid agar plates do not support sufficient growth for the microorganism inside the blood samples, automated blood culture systems were invented to ameliorate the sensitivity of this technique [17]. Blood samples obtained from the suspected patients are inoculated into the blood culture bottles containing enhanced broth media supporting sufficient growth of causative agents and resin or charcoal are added to counteract the effect of the antibiotic, allowing higher recovery of microorganisms inside the blood samples. Blood culture bottles are in incubated at 35 °C for five days and agitated to facilitate further growth of the microorganism [17]. Carbon dioxide generated by the growing microorganism is measured in either a fluorescent or colorimetric way [17]. Although adopting such technologies showed substantially increased sensitivity [18,19,20], the high costs associated with the device and blood culture bottles, regular instrument maintenance, and the infrastructural issues prevent their wide usage in low-income and developing countries [17]. As a result, manual culture systems are more popular in these regions. In manual blood culture methods, blood samples obtained from suspected patients are inoculated into either broth or biphasic blood bottles and incubated at 35 °C for five days in a static way [17]. Any visual changes, such as production of gas and turbidity, are measured by the naked eye [17]. Apart from the lower sensitivity of manual blood cultures when compared to automated ones, there are other complications when dealing with fungal agents inside the blood samples [17]. For instance, the vast majority of the blood bottles and the downstream plates used for the culturing of positive blood samples for fungal agents are mainly specific for bacterial species [5], while species of *Malassezia* in particular [21] and fungal species, in general, require modified media to increase the sensitivity rate of cultures. This notion along with the lower number of fungal cells in circulation when compared to bacterial ones, are the two main factors resulting in underestimation of fungal agents in clinical settings [5]. Future advances in the machinery and the chemical compositions of the blood culture media may not only make the automated device affordable for developing countries but may also enhance the universal issue of the low sensitivity of this technique.

### 2.2. Other Phenotypic Assays

Phenotypic assays such as Gram staining, direct microscopy, culture, and morphological assessment of isolates, combined with available biochemical assays are among the most popular and affordable identification tools used in laboratories of developing countries. These techniques can be applied either directly to clinical specimens (e.g., Gram staining) or to cultured isolates (e.g., morphological and biochemical evaluation).

Calcofluor white, a fluorescent dye, is used to easily detect fungal structures in clinical samples, but the required fluorescent microscopy may not be available to implement this analysis. Therefore, bright-field microscopy is generally a more feasible method and Gram staining is one of the simplest and least expensive differentiating techniques, which can be used to reveal fungal structures and host immune cells [22] in various sample types including tissue, cerebrospinal fluid, respiratory secretions, and vaginal discharge, etc. Potassium hydroxide is used to visualize fungal elements in dermal samples, whereas specialized stains such as Gomori methenamine silver (GMS), periodic acid-Schiff (PAS) stain, and hematoxylin and eosin (H&E) are more suitable for tissue samples [23]. Although these microscopic methods are not very discriminatory, they can at least provide a basis for prescribing empirical antifungal therapy.

Chromogenic agars can be used for the presumptive identification of some yeasts most commonly found in clinical samples [3], but only a few species, all belonging to the *Candida* genus, can be identified with these media, therefore, this approach is expensive and limited in scope. Microscopic observation of cultured isolates has high specificity for differentiation of *Candida albicans*/*C*. *dubliniensis* from other yeasts using the germ tube test, which is inexpensive but requires expertise in microscopy. Unfortunately, incorrect results are not uncommon [24]. Other highly specific tests to screen for *C*. *albicans*/*C*. *dubliniensis* or *C. glabrata* are based on detection of β-galactosaminidase or trehalase, respectively [24], but these may be cost-prohibitive for laboratories in developing countries. Commercially available manual biochemical methods such as API 20C AUX, rapid Yeast Plus, ID32C, etc., can provide a higher level of resolution for identification of cultures of many yeast and yeast-like species and may be used as an alternative. However, additional testing such as microscopic assessment of morphology on cornmeal- or rice-Tween 80 agars may be required to differentiate among biochemically similar species (Table 1). The review of Pincus et al. [24] can be consulted for comparing characteristics and performances of various manual biochemical methods. Still, none of the commercially available manual biochemical assays can be used to correctly identify challenging yeast species such as *C*. *auris* [25,26]. Furthermore, in some cases these methods may misidentify the top five *Candida* species [27]. Consequently, to ensure more accurate identification of fungal species in laboratories of developing countries that rely strongly on manual biochemical methods, either the current databases of commercially available biochemical assays should be updated/extended or these assays should be used in conjunction with other specific and sensitive molecular tools [27].

Additional phenotypic features that can help discriminate biochemically similar yeasts include temperature tolerance (growth at 37 °C or 42 °C), cell shape, pigment production, urease activity, KNO_3_ assimilation, etc. [24]. Table 2 shows the features of various phenotypic methods used for the identification of yeasts and yeast-like microorganisms.

Contrary to yeast species, identification of molds at the species level is either difficult or impossible without molecular methods such as MALDI-TOF MS or DNA sequencing and requires specific training in fungal morphology. Typically, mold identification is limited to a higher taxonomy level (group/complex or genus) and its precision depends on the skills of the technician conducting the morphological examination. Such an identification level is often sufficient for application of targeted therapy, with the exception of some cryptic species. Thus, *Aspergillus lentulus* should be identified to the species level rather than just to *Aspergillus* Section *Fumigati* (*Aspergillus fumigatus* complex), which would help avoid inappropriate therapy because this species has potential resistance to antifungals, e.g., amphotericin B, itraconazole, voriconazole, and caspofungin [28].

Highly specific methods such as MALDI-TOF MS require expensive equipment and are not generally affordable for laboratories in developing countries. However, significant savings can be achieved by improving the workflow and reducing reagent costs. In a large laboratory, cost savings per sample can offset the initial investment into equipment after about three years [29], although annual service contract fees can still be prohibitive. The higher level of specificity afforded by MALDI-TOF MS can have a positive impact on patient care, as it can help to avoid inappropriate therapies, such as over-treatment with fluconazole and ensure targeted therapy, which would decrease morbidity and mortality and reduce healthcare expenses associated with prolonged hospital stay.

## 3. Serological Assays

Serological tests were first used for the diagnosis of fungal pathogens in the early 1900s. By the 1950s, complement fixation and precipitin tests were introduced for serodiagnostics of coccidioidomycosis [30], followed by detection of the cryptococcal antigen as a serologically reactive substance in body fluids [31]. In 1958, Heiner adapted the Ouchterlony’s bacteriological agar-precipitin method for serodiagnostics of histoplasmosis [32] and in 1963, the first clinical application of serodiagnostics in medical mycology was reported when cryptococcal meningitis was diagnosed by antigen detection using a latex agglutination test [33].

These early findings led to the subsequent widespread use of antigen- and antibody-based detection of fungal pathogens. The diagnostic applications, which are described in detail below, include detection of antigens from the fungal cell wall (chitin, β-glucans, and mannoproteins), capsule constituents (glucuronoxylomannan), and metabolites (d-arabinitol and secreted aspartyl proteinase), as well as antibodies generated by the host immune system [34,35,36,37]. Although fungal culture and PCR are recommended for routine diagnostics, these methods are either time-consuming or require specific instrumentation, respectively. Therefore, serological testing plays an important role in screening for invasive fungal disease as it offers the following advantages: (i) rapid diagnosis, (ii) early initiation of antifungal therapy, and (iii) clinically significant information related to positive cultures (once contamination is ruled out) [36,37]. The most widely used clinical samples are bronchoalveolar lavage (BAL), cerebrospinal fluid, serum, whole blood, plasma, and urine. Detection of antigens is preferred over that of antibodies, as the latter may not be generated up to detectable levels due to the early stage of the infection or to immunosuppression [34,35,36,37].

### 3.1. Candidiasis

Invasive *Candida* infection is the most common form of invasive fungal disease. The 1,3-beta-d-glucan (BDG) test is known to produce false-positive results in certain risk groups and either fails to detect or provides only minimal detection of some fungal species, such as the zygomycetes, cryptococci, and *Blastomyces dermatitidis* [38], which limits its clinical value for screening purposes [39]. In contrast, combined testing of mannan, a polysaccharide that is non-covalently bound to the yeast cell wall and acts as an antigen, and anti-mannan antibodies (Mn/A-Mn) using ELISA is a useful and specific strategy to diagnose invasive candidiasis [40,41], which is employed in several clinical centers in developing countries.

### 3.2. Aspergillosis

*Aspergillus* antigens are an important biomarker for the early diagnosis of invasive aspergillosis (IA) [42]. In their landmark study of 1979, Reiss and Lehmann [43] found that galactomannan (GM) antigen, a polysaccharide present in the cell wall of most *Aspergillus* species, is released from growing hyphae during infection. An *Aspergillus*-specific lateral flow device (LFD) assay and the BDG test are also recognized as IA diagnostic tools [44,45]. The LFD assay utilizes the monoclonal antibody MAb JF5 for the detection of an extracellular glycoprotein secreted by *Aspergillus* during active growth [44]. Testing of BAL fluid has also improved the sensitivity and specificity of IA diagnosis [37]. A recent review indicates that positive results obtained in both GM and BDG tests or in the prototype LFD assay could confirm IA, whereas negative GM and LFD results reliably ruled it out [45]. These tests are currently available and applied in developing countries [46].

Recently, a newly formatted *Aspergillus*-specific LFD that provides a point-of-care test for the diagnosis of IA was CE marked and made commercially available by OLM Diagnostics [47,48,49]. The test has a good diagnostic performance for diagnosing IA in hematology patients [47,48,49] and non-neutropenic patients [50], but its low predictive value limits its value for a reliable diagnosis. The test is used in many developing countries but still requires clinical validation. An *Aspergillus* galactomannan LFD assay has also been introduced as a new point-of-care test for detection of galactomannan-like antigen, but this test requires some basic treatment in the laboratory. Nevertheless, it would be suitable for many developing countries [48,49,50]. The combined use of an *Aspergillus* galactomannan LFD assay and an *Aspergillus*-specific LFD has achieved 80% sensitivity and specificity for diagnosis of IA in non-neutropenic patients [50]. However, despite the important advances that have been made IA remains difficult to diagnose.

Elevated serum levels of *Aspergillus*-specific IgG and IgE are indicative of chronic pulmonary aspergillosis and allergic aspergillosis, which may be complicated by pulmonary tuberculosis resulting in death [51,52]. Importantly, serum levels of *A*. *fumigatus*-specific IgG and GM-specific IgG decrease inconsistently during the treatment of chronic pulmonary aspergillosis and may not be useful indicators for monitoring patient response [53]. These tests are also available in low-income [51] and middle-income [52] countries.

### 3.3. Cryptococcosis

Cryptococcal meningitis is diagnosed by detection of capsular polysaccharide glucuronoxylomannan in body fluids, including serum and cerebrospinal fluid. A well-known example of this diagnostic approach is the latex agglutination test widely used during the last four decades [54,55]. Recently, latex agglutination and enzyme-based immunoassay tests have been replaced by the LFD assay, which has similar or higher sensitivity, similar specificity, and lower cost [56,57,58,59]. The LFD assay also enables the detection of all *Cryptococcus* serotypes (A–D). This assay is inexpensive, user-friendly, provides a clear result within 15 min, and is equipment-free [54,60], which makes it well-suited for developing countries [60]. The World Health Organization (WHO) has also guided clinicians in the diagnosis and treatment of cryptococcosis and has recommended cryptococcal antigen testing in patients suffering from advanced HIV infection [61].

### 3.4. Mucormycosis

A novel MALDI-TOF MS assay based on detection of a panfungal disaccharide component in serum has been recently developed to diagnose invasive fungal disease [62]. This test has the advantage of detecting fungi of the *Mucorales* order that lack cell wall glucans. Unfortunately, this test is unavailable in low-income countries, which complicates the efficient diagnosis of mucormycosis in these regions.

### 3.5. Mycetoma

Mycetoma is a neglected tropical disease, which can be caused by bacteria (actinomycetoma) or fungi (eumycetoma) and for which no reliable diagnostic methods are currently available [63]. Among eumycetoma-causing pathogens, serological detection can be performed only for *Madurella mycetomatis* and *Pseudallescheria boydii* using counter immunoelectrophoresis, ELISA, or immunodiffusion. However, antigen preparation for these tests is cumbersome, not standardized, and can result in cross-reactivity [63].

### 3.6. Dimorphic Mycoses

Endemic systemic mycoses are life-threatening diseases which are challenging from both diagnostic and therapeutic points of view. Serodiagnostic methods have been improved for many dimorphic-diphasic fungi, including *Histoplasma capsulatum* [64], *Coccidioides immitis* [65], *Paracoccidioides brasiliensis* [66], *Blastomyces dermatitidis* [67], *Sporothrix schenckii* [68], and *Talaromyces marneffei* [69]. However, most of these techniques are not currently available in developing countries.

## 4. Molecular and PCR-Based Assays

### 4.1. Isothermal-Based Assays

The vast majority of isothermal amplification-based assays were developed around 1990, when PCR was just invented, and initially considered an expensive assay as it required a thermocycler. Consequently, various isothermal assays based on DNA template amplification at either room temperature or in a heating block with a constant temperature, have been developed with the aim to avoid the purchase of a costly instrument. Therefore, isothermal amplification assays could be a desirable inexpensive methodology in resource-limited countries and laboratories without access to PCR. In this section, we discuss the performance of isothermal assays widely used for identification of clinically important fungal species, including loop-mediated isothermal amplification (LAMP), rolling circling amplification (RCA), recombinant polymerase amplification (RPA), and nucleic acid sequence-based amplification (NASBA). Comparative diagnostic properties of these assays are summarized in Table 3.

#### 4.1.1. LAMP

This technique developed by a group of Japanese scientists in 2000 [70], is based on using four specific primers, two internal and two external ones, and a highly active displacing DNA polymerase enzyme yielding amplicons with a cauliflower-like structure [70]. In next-generation LAMP, which employs loop primers, the original LAMP amplification time was significantly reduced (up to 30 min) [71]. Rapidity (less than 1 h), high sensitivity (detects up to six target copies), resistance to inhibitors, and easy visualization provided by simple amplicon detection systems [72] make LAMP one of the most popular isothermal assays for identification of a wide range of clinically important fungi from pure culture [73] as well as from environmental [74] and clinical [75,76,77,78,79] samples. The technique has been successfully optimized and validated for detection of *C*. *auris* from simulated environmental samples and clinical specimens and in some cases, it demonstrated superiority over quantitative real-time PCR (qRT-PCR) for direct identification of *A*. *fumigatus* in clinical samples [75,78]. The main drawbacks of LAMP include complexity of primer design and assay optimization, inability to perform multiplex analysis, high sensitivity to carry-over contamination, and dependence on an additional heating block (95 °C) for increasing sensitivity.

#### 4.1.2. RCA

Mimicking bacterial plasmid amplification, this technology utilizes several primers that anneal to various sites in a circular DNA molecule, which allows rapid and efficient amplification [80]. RCA provides high efficiency and specificity, a high level of amplification, mutation detection, quantification, and versatility (use in both solid and liquid phases). Furthermore, it requires little to no optimization [72,81]. RCA-amplified products can be detected by gel electrophoresis and even colorimetric assays using DNA-intercalating dyes [82]. In medical mycology, RCA is mostly used with circularized (padlock) probes (RCA-padlock probes) to increase specificity and sensitivity. When two complementary ends of a probe match the template, they are juxtaposed and ligated by T4 ligase, then, RCA primers anneal and amplification occurs [83]. Applied to DNA isolated from pure cultures, RCA-padlock probes not only successfully differentiated among a wide range of medically important fungal species but also distinguished various genotypes of Cryptococcus and Trichophyton species [84,85,86,87,88,89,90,91,92,93,94,95]. However, although RCA is extensively applied to analyze DNA samples derived from pure cultures, only a few studies employed this technique for direct detection of pathogens in clinical samples and their results indicated poor sensitivity [84]. The use of semi-nested PCR prior to RCA can overcome this drawback [85]. Furthermore, carry-over contamination in negative control samples along with unusual banding patterns observed in some cases can be confusing [79]. Due to the low specificity of the method, it was suggested that RCA should not be used in clinical laboratories. Moreover, it was found that the specificity of RCA-padlock probes for detecting mismatches mostly relies on the nucleotide type [87].

#### 4.1.3. RPA

This approach, introduced in 2006, is based on using recombinase A (*RecA*), which unwinds the DNA molecule. The resulting unwound target region is then stabilized by single-stranded (ss)DNA-binding proteins and amplified by *Sau* polymerase [96,97]. With an amplification time of 15–30 min, RPA is the fastest among isothermal techniques. Availability of a wide range of commercial kits along with the similarity of primer design principles to conventional PCR are the other advantages of this technique [96]. RPA applied to specimens obtained from *M*. *mycetomatis*-infected patients showed 100% sensitivity and specificity [98]. However, as RPA is the newest among the isothermal approaches, it is still rarely applied in medical mycology.

#### 4.1.4. NASBA

Invented in 1991, this technique uses RNA as a template, a combination of two primers (one binding to RNA and the other to ssDNA), a T7 RNA polymerase, reverse transcriptase (Avian Myeloblastosis Virus [AMV]), and RNase H [99]. Among the isothermal techniques applied in medical mycology, NASBA proved to be a promising method to detect *Aspergillus* and *Candida* species in blood samples of patients suffering from IA and invasive candidiasis [100,101]. The method was also shown to be useful for predicting 12-week outcomes [100] and to have higher sensitivity than qRT-PCR in patients with IA [102,103]. A combination of NASBA and the BDG test with neutrophil count substantially increased the sensitivity of both techniques and specificity of NASBA [104,105]. Compared to qRT-PCR, NASBA requires smaller sample volume (only 100 µL of blood) [103] and less labor-intensive RNA extraction [101,103]. However, as RNA in blood samples is prone to degradation even at –70 °C, it should be extracted from fresh blood specimens treated with protective buffers [101,103].

### 4.2. PCR-Based Assays

Since its discovery and subsequent commercialization, PCR has gained vast popularity and has become an indispensable identification tool with a plethora of applications in medical microbiology. Owing to its affordability and reproducibility, PCR is recommended by the WHO as a reliable diagnostic tool to be used in developing countries [108]. Recent technological advances enabled the manufacturing of instruments that can use the most basic and off-the-shelf home appliances yet still demonstrate good performance [109]. PCR is a versatile technique used in a wide range of applications. The assays employed in medical mycology are based on conventional PCR-based assays (i.e., normal PCR, PCR-RFLP, and nested PCR) and qRT-PCR. Various properties of PCR-based assays are summarized in Table 4.

#### 4.2.1. Conventional PCR-Based Methods

Conventional PCR-based assays utilize two primers to amplify targets in singleplex or multiplex reactions with subsequent visualization by gel electrophoresis. Variations in the length and banding patterns of PCR products along with restriction digestion profiles enable differentiation among various target fungal species. Conventional PCR-based assays are categorized as simple PCR, restriction fragment length polymorphism (RFLP)-PCR, and nested-PCR.

##### PCR-RFLP Assays

This method is a combination of PCR and restriction digestion. PCR-amplified products obtained with sets of universal primers are checked by gel electrophoresis and digested by restriction endonucleases. The resulting specific fragmentation patterns visualized by gel electrophoresis are used for identification of various fungal species. PCR-RFLP is applied to analyze DNA obtained from both pure cultures and clinical samples and can detect a range of pathogenic fungi, including *Candida* species [110], *Histoplasma capsulatum* [110], *Mucorales* [111,112], and *Aspergillus* [113]. However, since PCR-RFLP consists of several steps, including electrophoretic separation and digestion with restriction enzymes, it carries the risk of carry-over contamination, requires additional expenses, and is time-consuming (in some cases, the identification process can take up to 48 h [110]). Moreover, if the obtained amplicons have multiple bands and/or similar lengths, the exact identification of species directly in clinical samples cannot be achieved during routine laboratory analysis.

##### Simple PCR

Simple PCR assays are relatively easier and more straightforward than PCR-RFLP assays and differentiation of target species mainly relies on specific primers and the length of the amplified fragments. Ease of primer design, straightforward processing, and low-cost account for the wide use of this assay in the identification of a broad range of clinically relevant fungal species, including the most common pathogenic yeasts [114], cryptic *Candida* species [115], onychomycoses agents [116], and *Aspergillus* [117,118]. Multiplicity significantly affects sensitivity and specificity resulting in a higher detection limit and non-specificity in some cases [119]. Although simple PCR does not have sufficient sensitivity to detect low numbers of target organisms and, therefore, may not provide desired results when applied directly to clinical specimens, it showed good results when used on blood cultures [120,121] or combined with reverse line blot hybridization, PCR-RLB [118]. Moreover, some of these multiplex PCR assays showed a higher degree of sensitivity when compared to phenotypic assays, such as direct microscopy and culture [116,118]. In addition to relatively low sensitivity, simple PCR assays depend on fragment visualization by gel electrophoresis, which can increase the turn-around time and the risk of carry-over contamination.

##### Nested PCR

The method is based on two consecutive PCR rounds. Fragments are amplified with outer primers in the first round and used as a template for inner primers in the second round. This strategy (two primer sets and 60–70 rounds of PCR runs) addresses the sensitivity and specificity issues of PCR-RFLP and simple PCR and is suitable for direct detection of low numbers of fungal cells in clinical samples, thus enabling timely administration of an appropriate antifungal [122]. Nested PCR can be used to identify *Aspergillus* and *Mucor* in paraffin wax-embedded tissues [123] and *Candida* species in blood samples [124,125]. Moreover, combined with RFLP, nested PCR showed a high degree of specificity and sensitivity for *Mucorales* species [112]. However, it should be noted that two consecutive PCR rounds can increase the chance of obtaining false-positive results (especially for fungi that occur ubiquitously in the environment) and augment turn-around time and expenses. For instance, detection of *Candida* species [124,125] and mucormycosis agents [112] by nested PCR from culture-negative specimens have been reported before, which could be associated with a lower sensitivity of culture [5], presence of transient candidemia episodes [124], and ability of PCR to capture non-viable cells [125]. These factors should be considered before applying nested PCR in clinical settings. Separation of PCR spaces (pre- and post-PCR), use of fungi-free reagents (buffers, enzymes) and specialized laboratory equipment (microfuge, pipettes, gloves, etc.), and strict adherence to appropriate protocols and instructions are of high importance when nested PCR assays are performed.

#### 4.2.2. Real-Time-PCR-Based Assays

Ease of application, low cost, and high specificity for the chosen targets demonstrated by conventional PCR, together with the increasing number of sequenced fungal genomes, which facilitates the design of specific primers, make this assay an important stand-alone diagnostic method in some parts of the world. However, the requirements for quantitative analysis and monitoring of amplification in real-time have led to the development of quantitative real-time PCR. Utilization of short amplicons (100–200 bp) resulting in time reduction, easy multiplexing due to availability of a wide range of reporters, recording and storage of quantitative data, and the high-throughput platform are the other advantages of qRT-PCR over conventional PCR.

In qRT-PCR, amplicons are detected using DNA-binding dyes and probes. SYBR Green is the most common DNA-binding dye, however, it can inhibit amplification at higher concentrations. Therefore, other dyes such as Eva Green, which is less inhibitory and produces more distinct melting curves, have been developed [131]. However, as both dyes bind any ds-DNA and, to a lesser degree, ss-DNA, there is a possibility of background signal. Therefore, fluorescent probes specific for the target DNA region of the amplicon are used in combination with specific primers, which significantly enhances specificity of qRT-PCR detection compared to reactions performed with DNA-binding dyes. Importantly, choice of target can greatly affect the diagnostic value using qRT-PCR, since targets within the ribosomal DNA locus are present in multiple copies, sometimes in excess of 100 copies per cell [132], in contrast to structural genes such as actin (*ACT*), elongation factor 1 alpha (*TEF1*α), RNA-polymerase II (*RPB2)* [133], which are present in single copy. However, even within the ribosomal locus, target choice can affect sensitivity and specificity. For example, White et al. [134], found the 28S subunit to be more sensitive and specific with a higher positive and negative predictive value than the 18S subunit for *Aspergillus fumigatus* detection. Taqman probes (ThermoFisher Scientific, Inc., Waltham, MA, USA) and Molecular Beacons (LGC, Bioresearch Technologies, Inc., Middlesex, UK) are among the most popular probes used for diagnostic purposes [135], however, there are extensive variations in probe chemistries and detection mechanisms, which can be selected based on a specific diagnostic need.

It should be noted that despite extensive application of different PCR-based detection methods in clinical practice in general, in diagnostic mycology their use is not yet so widespread since they have been slow to enter the market and to be approved by the Food and Drug Administration in the United States for a number of reasons [39]. Most PCR-based approaches target *Candida* spp. and their use for identification of the other fungi is still limited. One of the toughest challenges is DNA template preparation and standardization, which is due to the low number of target fungal cells present in circulation [5] as well as the need of a physical disruption step for cell breakage and another step to clean the template [136]. The choice of a clinical sample (whole blood, serum, plasma, pellet, supernatant, etc.) is also a factor as different fractions may have inhibitory compounds, which should be removed. Another problem is the difficulty of standardizing components required for PCR [137]. DNA target selection should also be considered, thus multicopy repeats such as ribosomal RNA genes offer better sensitivity and specificity for fungal identification in clinical samples [138].

Finally, because of the high sensitivity of PCR, particular care must be taken to perform reactions under clean conditions to prevent contamination and to include adequate controls. Template preparation and PCR setup require dedicated areas separated from that of electrophoresis to avoid contamination due to a large number of target copies present in a single band in the gel. PCR workstations, including a hood with light and UV decontamination, dedicated pipettors, barrier tips, and tubes are common in laboratories performing high throughput diagnostic assays. PCR assays need multiple controls for reagent contamination, internal amplification control for inhibitory substances, as well as negative and positive controls. Although these requirements can be challenging, various groups have been working to improve PCR for application in fungal diagnostics, including the International Society for Human and Animal Mycoses and the European *Aspergillus* PCR initiative. Through these efforts, more assays should enter clinical practice as approved methods for sensitive detection of fungal species in clinical samples.

## 5. Rapid Identification of Antifungal Resistance Using Molecular Techniques

Studies on susceptibility to antifungal agents revealed that resistance to azole compounds is predominantly caused by the acquisition of various universal and species-specific mutations in the genes encoding lanosterol-alpha demethylase (*ERG11*) in yeast species [139,140,141] and cytochrome P-51A (*cyp5A1*) in *A*. *fumigatus* [142]. Overexpression of efflux pumps and azole target genes is also a contributing factor. In addition, yeast species resistant to echinocandins have been reported to acquire resistance mutations in hotspot regions of 1,3-beta-glucan synthase gene components *FKS1* and/or *FKS2* [143]. In vitro phenotypic susceptibility testing requires pure cultures and can take 24–48 h to obtain results, whereas molecular assays can detect antifungal resistance directly in clinical samples, which can significantly shorten the turn-around time, thus enabling timely prescription of appropriate antifungal drugs. Moreover, rapid diagnostic assays can be employed to screen environmental samples for infection control purposes. LAMP [144] and nested and qRT-PCR [145,146] have been widely used for detection of azole-resistant *A*. *fumigatus* strains in pure cultures and clinical samples. Conventional and qRT-PCR assays have been developed to identify azole- and echinocandin-resistant *Candida* isolates [147,148,149]. However, species-specific mutations and regulatory networks conferring resistance to azoles [150] preclude the establishment of a comprehensive diagnostic assay that can detect 100% of azole-resistant fungal isolates. Identification of nonsynonymous mutations supplemented by X-ray crystallography and detailed transcriptomic studies comparing antifungal-resistant and -susceptible isolates may reveal key components to be used as biomarkers for the development of more comprehensive molecular assays.

## 6. Promising Future Technologies

### 6.1. CRISPR-Based Diagnostic Tools

To circumvent such limitations as low copy numbers of fungal DNA/RNA in human body fluids, lack of staff expertise, and the need of sensitive, specific, rapid, and inexpensive identification tools operating at room temperature, investigators searched for an approach that can be easily integrated into instrument-free environments in resource-limited countries. The versatility of the CRISPR-Cas9 (clustered regulatory interspersed short palindromic sequences-CRISPR associated protein 9) technology allowed the development of an identification tool called Specific Highly Sensitive Enzymatic Reporter UnLOCKING (SHERLOCK), which could successfully identify target nucleic acids in attomolar concentrations, differentiate closely-related viruses, and genotype single base-pair differences [151]. New-generation SHERLOCK version 2 represents a quantitative multiplex assay in which final results are visualized using an LFD system [152]. Combined with a fast DNA extraction protocol and the HUDSON (heating unextracted diagnostic samples to obliterate nucleases) method, SHERLOCK v2 demonstrated high sensitivity and specificity and could be successfully used as a point-of-care detection system for identification of Zika and dengue viruses [153]. Thus, SHERLOCK v2 supplemented with an efficient DNA extraction tool holds promise as a portable platform to identify pathogenic fungal species.

### 6.2. Nanopore Sequencing

The DNA sequencing of both genomes and specific targets has become an increasingly important diagnostic tool for clinical microbiologists. However, sequencing systems such as Sanger sequencing are out of reach for most small clinical laboratories in developing countries, with the rare exception of regional reference laboratories [154], because DNA sequencing equipment requires substantial laboratory spaces and significant capital investments. The recently introduced 4th generation sequencing technology has yielded multiple systems potentially suitable for developing countries, among them, nanopore sequencing is the latest tool entering the market.

Nanopore sequencing became possible after the first description of DNA translocation across a lipid bilayer membrane through a pore created by *Staphylococcus aureus* alpha-hemolysin upon application of an electric field [155]. This strategy was commercialized almost 20 years later by multiple companies, which have developed different platforms [5]. Two companies, Pacific Biosystems, Inc. (Menlo Park, CA, USA) <https://www.pacb.com/> and Oxford Nanopore Technologies (Oxford, UK) <https://nanoporetech.com/>, successfully released their products into the market. Oxford Nanopore Technologies has developed multiple devices with very small space requirements, including one the size of a computer memory stick called the MinION [156]. The small size allows the device to be run by a laptop computer and the data can be uploaded to the Cloud and analyzed with vendor software or directly by users with their own pipelines or programs. Compared to other sequencing instruments, MinION is much cheaper, costing about US$ 1000 for a starter pack which includes a sequencing module, flow cell, and sequencing reagents. Although the cost per run is presently more than that for Sanger sequencing (which can be a few dollars for a single sequence), MinION sequencing can be multiplexed either directly with one of the many sequencing kits or barcoded by the user with specific primers. There are add-on components such as a template extraction device (Voltrax), which can be docked with the sequencer, and a separate computer module (MinIT), which can replace the laptop.

Although the technology is very recent, it has already been applied to fungal detection in a number of studies. Ashikawa et al. [157] used nanopore sequencing to identify five species of *Candida* in positive blood culture bottles and compared its performance with Sanger sequencing. Wurzbacher et al. [158] used nanopore sequencing to screen fungal herbarium specimens for a rarely studied fungal ribosomal repeat (intergenic spacer, IGS), which is longer than the internal transcribed spacer (ITS), but it is not a good choice for Sanger sequencing due to the numerous reactions that are required to cover the length, time it takes to complete the multiple reactions, and costs. The MinION sequencer easily processes these regions, as this platform can sequence templates hundreds of kilobases long, and showed similar accuracy to Sanger sequencing [159]. For fungal identification, a database of IGS sequences similar to the ITS databases would be required for the strategy to be truly effective. However, if used for whole-genome sequencing, nanopore sequencing could potentially provide both identification and susceptibility data in a single run.

One of the most advantageous features of nanopore sequencing is rapid optimization of instrumentation and reagents. This system can be used to create true point-of-care devices, which could be easily deployed in the clinic or field and facilitate rapid and reliable diagnosis. In combination with companion platforms for automatic DNA template preparation, nanopore sequencing potentially enables sample-to-answer sequencing and can provide advanced molecular diagnostics in the most remote areas of the world without significant capital investments. The comprehensive review by Gabaldon et al. can be consulted to obtain more comprehensive information regarding diagnostic application of various molecular assays ranging from PCR to next-generation (NGS) platforms [160].

## 7. Conclusions

As the number of fungal species causing human infections increases, the lack of accurate and rapid diagnostic tools poses a challenge for epidemiological studies in resource-poor and developing countries, whose financial resources are limited to obtain expensive diagnostic instrumentation. To circumvent this problem, alternative affordable diagnostic/identification tools should be provided or grants and free services offered to encourage the installation of MALDI-TOF MS platforms in laboratories of these countries. On the other hand, although some medical mycological groups launched important initiatives to find solutions and presented meaningful and accurate data about invasive fungal infections, collaborations among developing countries should be extended in order to define diagnostic limitations and devise comprehensive strategies to improve diagnostics and treatment of fungal infections. Availability of inexpensive and portable sequencing machines applicable in the field along with the abundance of online sequencing data of mutations conferring resistance to azoles and echinocandins from online databases, such as www.theyeasts.org, may allow timely institution of appropriate antifungal drugs and avoiding expensive and time-consuming phenotypic antifungal susceptibility testing.

## Figures and Tables

**Table 1 jof-05-00090-t001:** Morphological differentiation of some yeast and yeast-like species similar in their biochemical reactivity profiles.

**Low Reactivity**	**PH ***	**TH**	**AR**	**CH**	**SP**
*Candida glabrata*	-	-	-	-	-
*Candida krusei*	+	-	-	-	-
*Candida lipolytica*	+	+	-	-	-
*Prototheca wickerhamii* (algae)	-	-	-	-	+
*Saprochaete capitata*	-	+	+	-	-
**Medium Reactivity**	**PH**	**TH**	**AR**	**CH**	**SP**
*Candida albicans*	+	+	-	+	-
*Candida lusitaniae*	+	-	-	-	-
*Candida parapsilosis*	+	-	-	-	-
*Candida tropicalis*	+	+	-	-	-
**High Reactivity**	**PH**	**TH**	**AR**	**CH**	**SP**
*Candida guilliermondii*	V	-	-	-	-
*Cryptococcus neoformans*	-	-	-	-	-
*Trichosporon asahii*	+	+	+	-	-

* PH, pseudohyphae; TH, true hyphae; AR, arthroconidia; CH, chlamydospores; SP, sporangiospores; +, present; -, absent; V, variable.

**Table 2 jof-05-00090-t002:** Diagnostic features of phenotypic methods useful for the identification of yeasts and yeast-like pathogens in developing countries [24].

Method	% Sensitivity	% Specificity	TAT	Reference
Rapid screening tests, e.g., β-galactosaminidase for identification of *Candida albicans*/*dubliniensis*	97.8–100.0	85.7–100.0	5–60 min	[24]
Rapid screening tests, e.g., trehalase for identification of *Candida glabrata*	89.3–100.0	74.1–100.0	30 s–24 h	[24]
Chromogenic agars, e.g., CHROMagar Candida, chromID Candida for identification of *Candida albicans*	88.3–100.0	86.0–100.0	48 h	[24]
Manual biochemical methods, e.g., api 20C AUX, ID32C, rapID Yeast Plus	N/A	86.0–100.0	4–72 h	[24]
Automated biochemical methods, e.g., MicroScan YID, VITEK 2 YST	N/A	85.3–98.5	4–48 h	[24]

TAT, turn-around time; N/A, not applicable. As the price for the phenotypic assays included are profoundly affected by the order scale and the tax amount, there is not a definitive price and it can vary from one country and one site to another.

**Table 3 jof-05-00090-t003:** Diagnostic properties of isothermal amplification-based techniques useful for cost-effective detection of fungal pathogens in developing countries.

Isothermal Approach	Sensitivity	Specificity	TAT	Quantification	Cost	Sample Source	Multiplexing	Easy Optimization	References
LAMP (conventional, intercalating dyes, and probes)	470 pg–0.2 fg	Controversial	120–60 min	No	3.2–5.3 Euros/rxn	Pure culture, clinical samples, simulated environmental and clinical samples	Yes	No/Yes	[70,71,73,74,75,76,77,78,79,106]
RCA-Padlock probes/RCA-Padlock probes+ Seminested-PCR	100 µg–40 fg ^A^ copies	Specific	300–120 min	No	2–5 USD/rxn	Pure culture, clinical samples, simulated environmental and clinical samples	No	No/Yes	[84,85,86,87,88,89,90,91,92,93,94,95]
RPA	230 pg	Specific	40 min	No	4.25 USD/rxn	Pure culture/clinical samples	No	No	[98]
NASBA	<10 fg ≤ 100 ag	Specific to minor CR	360–120 min	Yes	NI	Clinical samples, simulated blood sample, and samples from animal models	No	No	[100,101,102,103,104,105,107]

TAT, turn-around time; pg, picogram; fg, femtogram; ag, atomgram; NI, Not-indicated; rxn, reaction. ^A^ The optimal sensitivity of this technique was provided by the combination of RCA probes and semi-nested PCR. Considering that 10^5^ cells are equal to 1 ng, all copy number values were converted to weight units (g) for consistency.

**Table 4 jof-05-00090-t004:** Diagnostic properties of PCR-based techniques.

PCR-Based Approaches	Sensitivity	Specificity	TAT	Quantification	Cost	Sample Source	Multiplexing	Easy Optimization	References
**PCR-RFLP**	100 pg/µL	High	~8–48 h	No	NI	Culture, clinical samples	Yes	NI	[110,111,112,113]
Conventional PCR	High ^A^	High ^B^	~3–8 h ^C^	No	0.75–1 Euros ^D^	Culture, nail samples, positive blood culture bottles, simulated blood samples, clinical samples	Yes	Yes	[114,115,116,117,118,119,120,121]
Nested-PCR	0.1–150 fg	High ^D^	~6–24 h	No	NI	Culture, paraffin wax embedded tissues, and blood sample	Yes	NI	[112,122,123,124,125]
Real-time PCR	10 fg ^E^	High ^F^	1–2 h	Yes	NI ^G^	Culture, clinical samples, formalin-fixed paraffin-embedded specimens, environmental	Yes	Can be complicated if primer optimization and melt curves are performed as well as using absolute values for calibration due to the need for standard curve generation	[126,127,128,129,130]

TAT, turn-around time; pg, picogram; fg, femtogram; ag, atomgram; NI, Not-indicated. ^A,B^, Sensitivity and specificity was high except for minor cases. ^C^ Timing was not clearly stated in some studies, hence TAT was calculated based on the approximate DNA extraction and PCR stages. Except for [120] that found the specificity of 2.15 ± 0.25 CFU/mL, the rest of culture-dependent assays sensitivity considered when all the target species were correctly identified and specificity was defined when the assay did not cross-react with non-target species. ^D^, price ranges were only mentioned in reference 114 and 115. ^E^ Sensitivity can be affected by multiple factors including platform, chemistry, and perhaps most importantly, extraction method. ^F^ Specificity of real-time PCR can be greatly affected depending on whether just a dye is used to detect non-specific amplification (SYBR Green) or a specific probe is used. ^G^ Cost can be greatly affected by detection method. A probe is much more expensive than dyes and a commercial master mix is more expensive than mixing individual components separately.

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
