# Peer review of "Identification of Mycoses in Developing Countries"

_jof, 2019, doi:10.3390/jof5040090_

Round 1

Reviewer 1 Report

This review focuses on identification of Mycoses in developing countries. The review is interesting, however requires some more in depth analysis of up to data literature, and increasing focus on the multiple POC bedside tests that are now available, specifically for Aspergillosis and Cryptococcosis.

Specific comments:

Abstract: I dont see POC LFDs for Aspergillosis and Cryptococcosis even mentioned. This needs to be changed. I dont really think that PCR needs to be mentioned that prominently in the abstract, as it does not have a central role for diagnosing the most common fungal infections anyways. You could rather also focus more on Galactomannan and also mention the new (and second) Asprgillus LFD that detects an GM like antigen.

- Overall: There is too much focus on PCR assays that are not that relevant for making the diagnosis of the most common fungal infections anywhere, and particularly less relevant for developing countries. Would suggest to shorten that sections and instead really improve and update the sections on the two new Aspergillus LFDs (one is not even mentioned), and the CrAG LFD, which may really be extremely relevant for developing countries.

- Aspergillus LFD: Line 165-172: The Aspergillus-specific lateral flow device (LFD) is mentioned, however the results are outdated focusing on an old prototype version of the test. In the meantime a CE marked version of the test is commercially available, and used in many developing countries throughot the world. There is also plenty of new literature discussing the new version that needs to be discussed (many of the current references could be replaced, as they report on the non-commercial prototype version. For Lit see: https://www.ncbi.nlm.nih.gov/pubmed/30651395 AND https://www.ncbi.nlm.nih.gov/pubmed/29972764 AND https://www.journalofinfection.com/article/S0163-4453(18)30319-0/abstract AND https://onlinelibrary.wiley.com/doi/full/10.1111/myc.12881.

Some of the papers above also include the Aspergillus Galactomannan Lateral Flow Assay, which is a new POC test for detection of Galactomannan like antigen that is currently not discussed at all in the paper. Thsi test requires some basic pretreatment in the lab but would still be suitable for many developing countries.

Also of note authors mention that the Aspergillus-specific LFD requires pretreatment of samples which is NOT true for the vast majority of BAL samples (except those with blood in it), which can be tested natively with the test, meaning the test can be performed at bedside and without a lab. Please correct and discuss accordingly.

- CrAG LFD assay: this is so central for developing countries and needs a much more in depth discussion. Please see below for some of the relevant literature: 

Anderson DA, Crowe SM, Garcia M. Point-of-care testing. Curr HIV/AIDS Rep. 2011;8(1):31–7.[PubMed] [Google Scholar] Vijayan T, Chiller T, Klausner JD. Sensitivity and specificity of a new cryptococcal antigen lateral flow assay in serum and cerebrospinal fluid. MLO Med Lab Obs. 2013;45(3):16. [PMC free article] [PubMed] [Google Scholar] 40. Kozel TR, Bauman SK. CrAg lateral flow assay for cryptococcosis. Expert Opin Med Diagn. 2012;6(3):245–51. [PMC free article] [PubMed] [Google Scholar] Binnicker MJ, Jespersen DJ, Bestrom JE, Rollins LO. Comparison of four assays for the detection of cryptococcal antigen. Clin Vaccine Immunol. 2012;19(12):1988–90. [PMC free article] [PubMed] [Google Scholar] McMullan BJ, Halliday C, Sorrell TC, Judd D, Sleiman S, Marriott D, et al. Clinical utility of the cryptococcal antigen lateral flow assay in a diagnostic mycology laboratory. PLoS One. 2012;7(11):e49541. [PMC free article] [PubMed] [Google Scholar] Escandon P, Lizarazo J, Agudelo CI, Chiller T, Castaneda E. Evaluation of a rapid lateral flow immunoassay for the detection of cryptococcal antigen for the early diagnosis of cryptococcosis in HIV patients in Colombia. Med Mycol. 2013;51(7):765–8. [PubMed] [Google Scholar]  Lindsley MD, Mekha N, Baggett HC, Surinthong Y, Autthateinchai R, Sawatwong P, et al. Evaluation of a newly developed lateral flow immunoassay for the diagnosis of cryptococcosis. Clin Infect Dis. 2011;53(4):321–5. [PMC free article] [PubMed] [Google Scholar] Magambo KA, Kalluvya SE, Kapoor SW, Seni J, Chofle AA, Fitzgerald DW, et al. Utility of urine and serum lateral flow assays to determine the prevalence and predictors of cryptococcal antigenemia in HIV-positive outpatients beginning antiretroviral therapy in Mwanza, Tanzania. J Int AIDS Soc. 2014;17:19040.[PMC free article] [PubMed] [Google Scholar] Boulware DR, Rolfes MA, Rajasingham R, von Hohenberg M, Qin Z, Taseera K, et al. Multisite validation of cryptococcal antigen lateral flow assay and quantification by laser thermal contrast. Emerg Infect Dis. 2014;20(1):45– [PMC free article] [PubMed] [Google Scholar] Kabanda T, Siedner MJ, Klausner JD, Muzoora C, Boulware DR. Point-of-care diagnosis and prognostication of cryptococcal meningitis with the cryptococcal antigen lateral flow assay on cerebrospinal fluid. Clin Infect Dis. 2014;58(1):113–6. [PMC free article] [PubMed] [Google Scholar]  Jarvis JN, Percival A, Bauman S, Pelfrey J, Meintjes G, Williams GN, et al. Evaluation of a novel point-of-care cryptococcal antigen test on serum, plasma, and urine from patients with HIV-associated cryptococcal meningitis. Clin Infect Dis. 2011;53(10):1019–23. [PMC free article] [PubMed] [Google Scholar]

-Line 167: Typo "bronhoalveolar lavage"

Author Response

Reviewer 1

This review focuses on identification of Mycoses in developing countries. The review is interesting, however requires some more in depth analysis of up to data literature, and increasing focus on the multiple POC bedside tests that are now available, specifically for Aspergillosis and Cryptococcosis.

Response: Thanks a lot for taking your valuable time and providing such updated information and constructive comments.

Specific comments:

Abstract: I dont see POC LFDs for Aspergillosis and Cryptococcosis even mentioned. This needs to be changed. I dont really think that PCR needs to be mentioned that prominently in the abstract, as it does not have a central role for diagnosing the most common fungal infections anyways. You could rather also focus more on Galactomannan and also mention the new (and second) Asprgillus LFD that detects an GM like antigen.

Response: As kindly requested by the reviewer, Abstract section has been modified accordingly.

Q1- Overall: There is too much focus on PCR assays that are not that relevant for making the diagnosis of the most common fungal infections anywhere, and particularly less relevant for developing countries. Would suggest to shorten that sections and instead really improve and update the sections on the two new Aspergillus LFDs (one is not even mentioned), and the CrAG LFD, which may really be extremely relevant for developing countries.

Q1 response. Thanks indeed for this valuable comment. We agree that the PCR section is a bit more extensive, which is due to the fact that the preliminary title of our paper was molecular identification of mycoses in developing countries. However, to avoid jumping into the molecular techniques we would like to give information regarding the background and utility of the other assays and the competence of laboratories of developing countries. Therefore, we suggested to change the title and included other assays to give the readers a general view. Moreover, although PCR assays are not well-established for diagnosis (which is a global issue yet to be solved and standardized), this technique is widely used for identification purposes.

Q2- Aspergillus LFD: Line 165-172: The Aspergillus-specific lateral flow device (LFD) is mentioned, however the results are outdated focusing on an old prototype version of the test. In the meantime a CE marked version of the test is commercially available, and used in many developing countries throughot the world. There is also plenty of new literature discussing the new version that needs to be discussed (many of the current references could be replaced, as they report on the non-commercial prototype version. For Lit see: https://www.ncbi.nlm.nih.gov/pubmed/30651395 AND https://www.ncbi.nlm.nih.gov/pubmed/29972764 AND https://www.journalofinfection.com/article/S0163-4453(18)30319-0/abstract AND https://onlinelibrary.wiley.com/doi/full/10.1111/myc.12881.

Q2 response. Thanks for your positive and informative comment. We revised the text according to your suggestion.

Q3- Some of the papers above also include the Aspergillus Galactomannan Lateral Flow Assay, which is a new POC test for detection of Galactomannan like antigen that is currently not discussed at all in the paper. Thsi test requires some basic pretreatment in the lab but would still be suitable for many developing countries.

Q3 response. Thanks for your positive and informative comment. We revised the text according to your suggestion.

Q4- Also of note authors mention that the Aspergillus-specific LFD requires pretreatment of samples which is NOT true for the vast majority of BAL samples (except those with blood in it), which can be tested natively with the test, meaning the test can be performed at bedside and without a lab. Please correct and discuss accordingly.

Q4 response. Thanks for your positive and informative comment. We revised the text according to your suggestion.

Q5- CrAG LFD assay: this is so central for developing countries and needs a much more in depth discussion. Please see below for some of the relevant literature:

Q5 response. Thanks for your positive and informative comment. We revised the text according to your suggestion.

Q6-Line 167: Typo "bronhoalveolar lavage"

Q6 response. Thanks for your kind comment. We revised the text according to your suggestion.

Reviewer 2 Report

My main concern with this paper is that the authors listed a large number of lab tests with almost no criticism in terms of the time they take to generate results, database available for accurate identification, suitability for routine use, costs, and accuracy. Consequently, the review paper adds very limited information to the field. I would not encourage the publication of a large list of tests without a more deeply discussion on their main advantages and limitations. Other aspect to be mentioned is that the authors put together platforms used for fungal detection and fungal identification along their discussion. Definitely,  paper requires a major review to become more interesting and useful for the readers.

My suggestions

Please, divide your paper in the following topics 1- Fungal detection and 2- Fungal identification

1-Fungal Detection- here you may discuss antigen tests, specific antibodies tests, automated systems for culturing, PCR based and other molecular assays available for detecting fungal infections in different biologic samples of the patients under investigation

2-Fungal Identification (ID)- here you may present methods used for fungal identification in positive cultures based on different strategies including screening with chromogenic media or PNA-FISH, conventional phenotypic tests for ID, MALDI-TOF and PCR based methods for ID

Details and criticism about all systems they they mentioned may be presented along comparative tables, illustrating the main characteristics and limitations of all methods.

To make easier the life of the readers, I would recommend to divide the presentation of all phenotypic methods for fungal ID as bellow

1-Screening with chromogenic media = Please, make comments on several chromogenic media available able to discriminate C albicans from non-albicans species. Other screening methods may also be included here

2- Manual biochemical methods- there are several commercial methods based on different biochemical tests that will be interpreted by a database that may contain very few species or a larger number of them. Please, inform what we may expect from them in a routine of lab, informing the major errors usually documented with these systems

3- Automated biochemical methods, like Vitek , Microscan, others. They should be presented in more details, in order to provide useful information for our readers. Please, inform what we may expect from them in a routine of lab, informing the major errors usually documented with these systems

Same strategy of discussion are valid for all different systems available for MALDI-TOF, PCR-based and other molecular assays used for fungal identification.

A final comment to the authors is that they should present some criticism about the public database available for identification of fungi by proteomics and sequencing of DNA targets considered to be informative for species identification.

Author Response

Reviewer 2,

Dear reviewer 2,

Thank you very much for devoting your precious time and providing valuable comments.

My main concern with this paper is that the authors listed a large number of lab tests with almost no criticism in terms of the time they take to generate results, database available for accurate identification, suitability for routine use, costs, and accuracy. Consequently, the review paper adds very limited information to the field. I would not encourage the publication of a large list of tests without a more deeply discussion on their main advantages and limitations. Other aspect to be mentioned is that the authors put together platforms used for fungal detection and fungal identification along their discussion. Definitely, paper requires a major review to become more interesting and useful for the readers.

My suggestions

Q1- Please, divide your paper in the following topics 1- Fungal detection and 2- Fungal identification

1-Fungal Detection- here you may discuss antigen tests, specific antibodies tests, automated systems for culturing, PCR based and other molecular assays available for detecting fungal infections in different biologic samples of the patients under investigation.

Q1 response: Thank you for making this valuable and important suggestion. The text has been revised accordingly and updated information has been inserted in the manuscript.

2-Fungal Identification (ID)- here you may present methods used for fungal identification in positive cultures based on different strategies including screening with chromogenic media or PNA-FISH, conventional phenotypic tests for ID, MALDI-TOF and PCR based methods for ID

Q2 response. Thanks for your comment. Please note that techniques, such as PNA-FISH, MALDI-TOF MS, and Sanger sequencing are not affordable for developing countries, which is highlighted in the introduction. According to your suggestion we divided the techniques discussed into Identification- and Detection-based assays. However, as this justification resulted in losing the integrity of the techniques discussed, all authors were unanimously content about the previous order. For instance, if you kindly see Table 4, you will see that even using conventional PCR, some studies found reasonable efficacy when applied on clinical samples. In the other words, not only conventional PCR but also other techniques, such as PCR-RFLP, and the isothermal-based techniques, have been used as both diagnostic and identification tools and dividing them into two sections will result in repetitive discussion and losing the integrity of the content. Therefore, we have uploaded two versions of the manuscript one arranged in a way suggested by reviewer 2 and the other one was according to the previous order. As for automated culture techniques they are not widely used in developing countries, which is mostly due to the high prices for blood bottles, the high price device purchase, maintenance, and finally infrastructural issue, whereas the vast majority of the hospitals in developing countries use the manual blood culture techniques. Therefore, we added another section dealing with the blood bottle cultures sorts, their description, and their utility in developing countries.

Q3- Details and criticism about all systems they they mentioned may be presented along comparative tables, illustrating the main characteristics and limitations of all methods.

Q2 response. This is indeed a great comment and we agree with it. According to your comment we used informative tables containing relevant diagnostic and identification information. Please note that we did not provide a comparative table for serology section, as the latest papers published in 2019, have extensively discussed and compared various serology assays and to prevent the repetition, we recommended readers to consult those papers for more information.

Q4- To make easier the life of the readers, I would recommend to divide the presentation of all phenotypic methods for fungal ID as bellow

1-Screening with chromogenic media = Please, make comments on several chromogenic media available able to discriminate C albicans from non-albicans species. Other screening methods may also be included here.

Q4 response. Please kindly find Table 2, where various properties, including sensitivity, specificity, and turn-around time are included and compared. Please note that there is no fixed price for biochemical assays and the final prices may vary significantly depending on the country and even site of use and the tax issued by the government. Hence, to prevent misinforming, we did not include the price range in this table and explained this in the footnote of the respective table.

Q5- Manual biochemical methods- there are several commercial methods based on different biochemical tests that will be interpreted by a database that may contain very few species or a larger number of them. Please, inform what we may expect from them in a routine of lab, informing the major errors usually documented with these systems

Q5 response. Please kindly find Table 2, where various properties, including sensitivity, specificity, and turn-around time are included and compared.

Q6- Automated biochemical methods, like Vitek , Microscan, others. They should be presented in more details, in order to provide useful information for our readers. Please, inform what we may expect from them in a routine of lab, informing the major errors usually documented with these systems

Q6 response. Please kindly find Table 2, where various properties, including sensitivity, specificity, and turn-around time are included and compared.

Q7- Same strategy of discussion are valid for all different systems available for MALDI-TOF, PCR-based and other molecular assays used for fungal identification.

Q7 response. Please find Tables 2 and 4 prepared in the way suggested by reviewer. Please note that MALDI-TOF MS, Sanger sequencing, and other expensive molecular assays, such as PNA-FISH, are cost-prohibitive to be established in developing countries. Therefore, they are out of scope of our study.

Q8- A final comment to the authors is that they should present some criticism about the public database available for identification of fungi by proteomics and sequencing of DNA targets considered to be informative for species identification.

Q8 response. Please note that MALDI-TOF MS, Sanger sequencing, and other expensive molecular assays, such as PNA-FISH, are cost-prohibitive to be established in developing countries. Therefore, they are out of scope of our study.

Round 2

Reviewer 1 Report

This manuscript is now ready for publication

Reviewer 2 Report

I consider that most suggestions were incorporated into the manuscript.